# Efficacy of mefloquine and its enantiomers in a murine model of *Mycobacterium avium* infection

**Antoine Froment**[1,2], **Julia Delomez**[1,3], **Sophie Da Nascimento**[1], **Alexandra Dassonville-Klimpt**[1], **Claire Andréjak**[1,3], **François Peltier**[1], **Cédric Joseph**[2], **Pascal Sonnet**[1], **Jean-Philippe Lanoix**[1,2]*

**1** AGIR UR-4294, Université de Picardie Jules Verne, Amiens, France, **2** Infectious Disease Department, Amiens-Picardie University Hospital, Amiens, France, **3** Pneumology Department, Amiens-Picardie University Hospital, Amiens, France

* lanoix.jean-philippe@chu-amiens.fr

**Data Availability Statement:** All relevant data are within the manuscript and its Supporting information files.

## Abstract

The treatment of *Mycobacterium avium* infections is still long, complex, and often poorly tolerated, besides emergence of resistances. New active molecules that are more effective and better tolerated are deeply needed. Mefloquine and its enantiomers ((+) Erythro-mefloquine ((+)-EMQ) and (-)-Erythro-mefloquine ((-)-EMQ)) have shown efficacy in both *in vitro* and *in vivo*, in a mouse model of *M. avium* intraveinous infection. However, no study reports aerosol model of infection or combination with gold standard treatment. That was the aim of our study. In an aerosol model of *M. avium* infection in BALB/c mice, we used five treatment groups as followed: Clarithromycin-Ethambutol-Rifampicin (CLR-EMB-RIF, standard of care, n = 15), CLR-EMB-MFQ (n = 15), CLR-EMB-(+)-EMQ (n = 15), CLR-EMB-(-)-EMQ (n = 15) and an untreated group (n = 25). To evaluate drug efficacy, we sacrificed each month over 3 months, 5 mice from each group. Lung homogenates were diluted and plated for colony forming unit count (CFU) expressed in Log10. At each time point, we found a significant difference between the untreated group and each of the treatment groups (p<0.005). The (+)-EMQ-CLR-EMB group was the group with the lowest CFU count at each time point but never reached statistical significance. The results of each group 3 months after treatment are: (+)-EMQ-CLR-EMB (4.43 ± 0.26), RIF-CLR-EMB (4.83 ± 0.37), (-)-EMQ-CLR-EMB (4.82 ± 0.18), MFQ-CLR-EMB (4.70 ± 0.21). In conclusion, MFQ and its enantiomers appear to be as effective as rifampicin in combination therapy. Further studies are needed to evaluate the ability of these drugs to prevent selection of clarithromycin resistant strains and potential for lung sterilization.

## Introduction

*Mycobacterium avium* infection is the most common non-tuberculous mycobacterial (NTM) infection globally, yet its treatment remains long, complex, and poorly tolerated [1]. The gold standard treatment involves Clarithromycin (CLR), Ethambutol (EMB), and Rifampin (RIF)

**Funding:** CA - Comité Départemental De Lutte Contre Les Maladies Respiratoires, an independent association fighting for patient's health. The funders had no role in study design, data collection and analysis, decision to publish, or preparation of the manuscript.

**Competing interests:** The authors have declared that no competing interests exist.

[2, 3]. However, resistance to both standard and alternative antibiotics is emerging [4–6]. New and repurposed antibiotics like clofazimine and mefloquine (MFQ) are urgently needed to potentially shorten treatment duration and improve tolerability.

Animal models are crucial for selecting combinations for clinical trials, with murine models of lung infection showing promise for bactericidal effects of clarithromycin and other drug combinations [7–10]. The aerosol model, in particular, mimics the human infection route and is well-suited for studying *M. avium* complex pulmonary disease (MAC-PD) [8, 11].

Mefloquine (MFQ), a racemic mixture of (+)-Erythro-mefloquine ((+)-EMQ) and (-)-Erythro-mefloquine ((-)-EMQ), has shown activity against *M. avium* by inhibiting DNA-synthase and mycolic acid transporter MmpL3 [12–15]. Prior studies using murine models of intravenous infection demonstrated its efficacy but did not explore combination therapies or compare enantiomers directly [16, 17].

This study aims to assess the efficacy of MFQ and its enantiomers ((+)-EMQ and (-)-EMQ) against the gold standard regimen in an aerosol model of *M. avium* pulmonary infections.

## Material and methods

### Mouse strain

BALB/c female 6-week-old mice were purchased from Charles River, Saint-Germain Nuelles, France.

### Bacterial strain

*M. avium* strain Chester (MAC 101; American Type Culture Collection (ATCC) 700898) was prepared as the inoculum after being grown to a log-phase culture in Middlebrook 7H9 broth (Difco, Detroit, MI) supplemented with 10% (vol/vol) oleic acid-albumin-dextrose (OADC; Becton Dickinson, Le Pont de Claix, France) and 0.05% (vol/vol) Tween 80 (Sigma, St. Louis, MO). *M. avium* cultures were incubated for 4 weeks at 37˚C before use in an experiment or infection. This strain was mouse-passaged from our previous experiment [7] and maintained frozen in -80˚C aliquots.

### Antibiotics

RIF was purchased from Sigma-Aldrich (Saint-Quentin Fallavier, France), dissolved in distilled water after grinding to obtain a concentration of 2mg/mL. It was administered by gavage 1 hour before other treatments [18]. EMB was purchased from Sigma-Aldrich, dissolved in distilled water to achieve a concentration of 20mg/mL. CLR was purchased from Arrow (Lyon, France) and from week 4 onwards from EG Labo (Issy-les-Moulineaux, France). CLR was ground and dissolved in 10% ethanol followed by distilled water to obtain a concentration of 20mg/mL. MFQ was purchased from Alfa chemistry and the enantiomers were separated by HPLC method using a Chiralpak IA column and heptane/isopropanol/diethylamine (70:30:0.1) as mobile phase [12]. Enantiomeric purity of the two enantiomers was determined after purification, assessed by a column chromatography technique and the enantiomeric excess of the (-)-EMQ and (+)-EMQ was of 79% and 95%, respectively. For solubility purposes, these molecules were dissolved in 10% ethanol, followed by distilled water to obtain a concentration of 8mg/mL.

### Ethics

All animal procedures were approved by the Animal Care and Use Committee of Amiens Picardy Jules Verne University: APAFIS#28969–2021011412489954 v2.

**Table 1. Experimental protocol.**

| Regimen | D-27 | D0 | M1 | M2 | M3 |
|---|---|---|---|---|---|
| Untreated (UT) | 5 | 5 | 5 | 5 | 5 |
| RIF+CLR+EMB | | | 5 *(4)* | 5 | 5 |
| MFQ+CLR+EMB | | | 5 | 5 | 5 |
| (+)-EMQ+CLR+EMB | | | 5 *(4)* | 5 | 5 |
| (-)-EMQ+CLR+EMB | | | 5 | 5 | 5 |
| Total = 85 *(83)* | 5 | 5 | 25 *(23)* | 25 | 25 |

Plain numbers are the number of mice planned for analysis. Numbers in italic and parenthesis are the number of mice actually used for the analysis. One mouse died from occlusion and one from polymicrobial axillary abscess

## Animal welfare

Mice were kept in collective cages (with a limit of 10 mice per cage) and enriched (rich, absorbent litter, tunnels, etc.). Food and drink were provided ad libitum. Animal welfare was evaluated at least once daily following Morton & Griffith score (see S1 File) and animals were weighted once a week. Humane endpoint were defined as respiratory trouble after gavage lasting more than 2 minutes in spite of Buprenorphine, bleeding or pain trouble after gavage lasting more than 5 minutes in spite of Buprenorphine, Morton & Griffith score >2 lasting more than 48h without improvement in spite of Buprenorphine or systemically if >7, loss of weight >20%. Sacrifices were performed by cervical dislocation. As required in the French law, all members of the staff in contact with mice hold a diploma in animal welfare.

## Experimental protocol (see Table 1)

Eighty-five mice were infected by aerosolization using an inhalation exposure system (Glas-Col, Terre Haute, IN) with 10 ml of log-phase culture of the Chester strain of *M. avium*. After aerosol infection, mice were randomized into 5 groups: Untreated (n = 25), CLR-EMB-RIF (n = 15), CLR-EMB-MFQ (n = 15), CLR-EMB-(+)-EMQ (n = 15), CLR-EMB-(-)-EMQ (n = 15). One day after infection, 5 mice were killed to assess bacterial implantation in lungs. Treatment was initiated 28 days after infection, defined as Day 0 (D0) for a duration of 3 months. At D0, 5 mice were killed to confirm infection progression. Finally, 5 mice per group were sacrificed at 1, 2 and 3 months. Daily doses were 10mg/kg for Rifampicin, 100mg/kg for Clarithromycin and Ethambutol and 40mg/kg for MFQ and its enantiomers. Each antibiotic was administered at a volume of 0.1mL per day by gavage, 5 days out of 7. Untreated mice did not receive either ethanol or water.

## Efficacy evaluation

Efficacy was assessed by CFU count obtained after culturing lung homogenates diluted in 2.5ml PBS (phosphate-buffered saline) solution. Serial dilutions were then made before plating on 7H11 medium (Millipore, St Quentin Fallavier, France) supplemented with 10% OADC and antimicrobials: 50mg/L Cycloheximidine, 50mg/L Carbenicillin, 25mg/L Polymyxin, and 20mg/L Trimethoprim. All these molecules were purchased from Sigma-Aldrich. CFUs were counted after 4 and 8 weeks of incubation at 37°C.

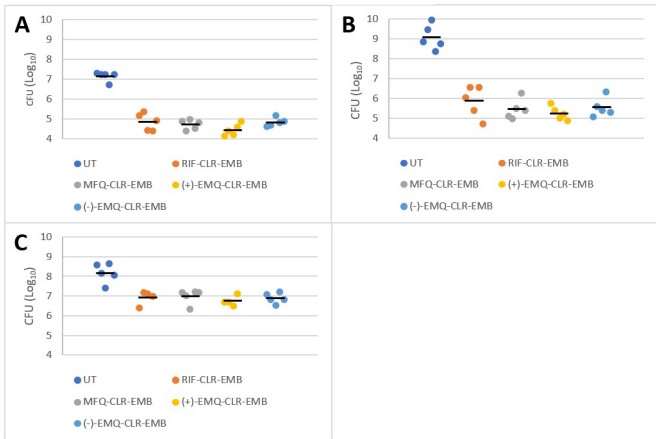

**Fig 1. Graphical representation of the results for each month.** Results of CFU counts after 8 weeks of culture on the two lungs homogenates. All values are represented by a dot, black horizontal bar is the mean for the group **A**. Dot plot representing results of M1 for treated **B**. Dot plot representing results of M2 for treated groups **C**. Dot plot representing results of M3 for treated groups.

## Statistical analysis

CFU counts (x) were transformed to log10(x+1) before analysis. To compare the effects of combinations, a one-way analysis of variance assuming a normal distribution was performed. A p-value of 0.05 was considered significant. The number of mice required was estimated at 5 per group to demonstrate a difference of 0.5 Log10 with a standard deviation of 0.3.

## Results

Mice were infected with a solution containing 10.2 log/ml of CFUs. At D-27 (one day after infection), the mean (± Standard deviation (SD)) CFU in lung was 6.24±0.14 log10 CFU, increasing to 6.68±0.19 at D0 before the first antibiotic administration.

One month after treatment (Fig 1A), the mean (+/- SD) CFU in each group was as follow: 6.91 ± 0.37 for RIF-CLR-EMB, 7.00 ± 0.36 for MFQ-CLR-EMB, 6.75 ± 0.26 for (+)-EMQ-CLR-EMB, 6.90 ± 0.26 for (-)-EMQ-CLR-EMB and 8.17 ± 0.49 for the untreated group. The difference between treated and untreated animals was statistically significant (p<0.005), however, there was no significant difference between the different treatment groups.

Two months after treatment (Fig 1B), the mean (+/- SD) CFU in each group was as follow: 5.87 ± 0.81 for RIF-CLR-EMB, 5.46 ± 0.51 for MFQ-CLR-EMB, 5.25 ± 0.36 for (+)-EMQ-CLR-EMB, 5.61 ± 0.52 for (-)-EMQ-CLR-EMB and 9.08 ± 0.63 for the untreated group. There were no significant differences between the treatment groups.

Three months after treatment (Fig 1C), the mean (+/- SD) CFU in each group was as follow: 4.84 ± 0.43 for RIF-CLR-EMB, 4.72 ± 0.25 for MFQ-CLR-EMB, 4.44 ± 0.29 for (+)-EMQ-CLR-EMB, 4.83 ± 0.20 for (-)-EMQ-CLR-EMB and 7.15 ± 0.25 for the untreated group, but there was no difference between the treated groups (p = 0.17). Whereas the CFU counts slightly increased at this time point in all treated animals, the difference with the untreated group persisted significantly. We represented the values below in terms of the difference between the treated and untreated groups (see Table 2).

**Table 2. Difference of CFU count between untreated group and treated groups.**

| Month | RIF-CLR-EMB | MFQ-CLR-EMB | (+)-EMQ-CLR-EMB | (-)-EMQ-CLR-EMB |
|---|---|---|---|---|
| 1 | 1.26 ± 0.29 | 1.18 ± 0.27 | 1.42 ± 0.26 | 1.28 ± 0.25 |
| 2 | 3.21 ± 0.46 | 3.62 ± 0.36 | 3.83 ± 0.32 | 3.47 ± 0.37 |
| 3 | 2.30 ± 0.22 | 2.42 ± 0.16 | 2.71 ± 0.17 | 2.32 ± 0.14 |

All values are represented in Log10 ± SD

## Discussion

To our knowledge, our study is the first to compare MFQ and its enantiomers with the gold standard treatment for MAC-PD in a combination of molecules. We found no significant differences between the standard regimen and MFQ-based combinations, suggesting that MFQ and its enantiomers could serve as potential replacements for RIF in MAC-PD treatment.

We focused on studying (+)-EMQ and (-)-EMQ enantiomers due to weaker in vitro activity of (+)-Threo-mefloquine and (-)-Threo-mefloquine [17]. Our objective was to evaluate RIF replacement given its well-known severe adverse effects, particularly hepatic and hematologic, which can lead to treatment discontinuation [19, 20]. Additionally, RIF is a powerful CYP3A4 inducer, causing numerous drug interactions [21, 22]. Although MFQ has significant neuro-psychiatric side effects, it causes fewer drug interactions, making it a less complex option for clinicians, especially in cases of RIF intolerance or resistance [12, 23, 24].

Despite the lower purity of (-)-EMQ, our results were consistent with previously reported efficacy data [17].

Our study has several limitations. Firstly, during the first month, two mice were euthanized according to humane endpoints, due to respiratory distress following gavage.

Secondly, an unexplained lack of bacterial growth on the third-month cultures, irrespective of treatment arms, necessitated a new plating using lung homogenates (stored for four weeks at 4˚C). This led to an underestimation of the initial number of colonies and an overestimation of treatment efficacy. Thus, results are presented as differences between untreated and treated groups, ensuring no impact on inter-group comparisons for a given time point.

Thirdly, our study was probably underpowered. Indeed, we expected 0.3 inter-individual variation and obtained 0.35, thus 10 mice per group would have allowed us to see a 0.4 log10 CFU difference. This higher observed variability might be explained by the use of a mouse-passaged MAC strain.

Finally, extending the study duration might reveal clearer differences.

In conclusion, our study showed no significant differences between the treatment groups, showing a potential for MFQ as RIF replacement. Further studies are necessary to evaluate MFQ's protective ability against the selection of clarithromycin resistance and its potential for lung sterilization.

## Supporting information

**S1 File.**
(XLSX)

## Acknowledgments

We thank PlatAnN members for the care provided to the animals of the study.

## Author Contributions

**Conceptualization:** Alexandra Dassonville-Klimpt, Claire Andréjak, Pascal Sonnet, Jean-Philippe Lanoix.

**Data curation:** Antoine Froment, Julia Delomez.

**Formal analysis:** Antoine Froment.

**Funding acquisition:** Claire Andréjak.

**Investigation:** Antoine Froment, Julia Delomez, Sophie Da Nascimento, Alexandra Dassonville-Klimpt, François Peltier, Cédric Joseph, Jean-Philippe Lanoix.

**Methodology:** Antoine Froment, Julia Delomez, Alexandra Dassonville-Klimpt, Jean-Philippe Lanoix.

**Project administration:** Pascal Sonnet, Jean-Philippe Lanoix.

**Resources:** Claire Andréjak, François Peltier, Jean-Philippe Lanoix.

**Software:** Antoine Froment, Julia Delomez, François Peltier, Jean-Philippe Lanoix.

**Supervision:** Pascal Sonnet, Jean-Philippe Lanoix.

**Validation:** Antoine Froment, Julia Delomez, Sophie Da Nascimento, Alexandra Dassonville-Klimpt, Claire Andréjak, Pascal Sonnet, Jean-Philippe Lanoix.

**Visualization:** Antoine Froment, Jean-Philippe Lanoix.

**Writing – original draft:** Antoine Froment.

**Writing – review & editing:** Antoine Froment, Claire Andréjak, Cédric Joseph, Pascal Sonnet, Jean-Philippe Lanoix.

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
