## [Decision Letter · Decision Letter 0]

9 Apr 2024

PONE-D-24-03260Efficacy of Mefloquine and its enantiomers in a murine model of Mycobacterium avium infectionPLOS ONE

Dear Dr. Froment,

Thank you for submitting your manuscript to PLOS ONE. After careful consideration, we feel that it has merit but does not fully meet PLOS ONE’s publication criteria as it currently stands. Therefore, we invite you to submit a revised version of the manuscript that addresses the points raised during the review process.

We look forward to receiving your revised manuscript.

Kind regards,

Delphi Chatterjee

Academic Editor

PLOS ONE

Journal Requirements:

1. When submitting your revision, we need you to address these additional requirements. Please ensure that your manuscript meets PLOS ONE's style requirements, including those for file naming. The PLOS ONE style templates can be found at https://journals.plos.org/plosone/s/file?id=wjVg/PLOSOne_formatting_sample_main_body.pdf and https://journals.plos.org/plosone/s/file?id=ba62/PLOSOne_formatting_sample_title_authors_affiliations.pdf 2. Thank you for your submission to PLOS ONE. We note that your study design may include death of a regulated animal as a likely outcome or planned experimental endpoint. At this time, we request that you please report additional details in your Methods section regarding animal care and use for the survival study, as per our editorial guidelines (http://journals.plos.org/plosone/s/submission-guidelines#loc-humane-endpoints).       For easy reference, we have attached a checklist that may be relevant for your submission. Please complete all items on the checklist at the following link:   http://journals.plos.org/plosone/s/file?id=bb1d/plos-one-humane-endpoints-checklist.docx          Please upload the completed checklist as file type “Other” when resubmitting your manuscript. This document is for internal journal use only and will not be published if your article is accepted. We very much appreciate your attention to these requests and support of improved reporting standards in PLOS ONE submissions. 3. To comply with PLOS ONE submissions requirements, in your Methods section, please provide additional information regarding the experiments involving animals and ensure you have included details on (a) methods of sacrifice, (b) methods of anesthesia and/or analgesia, and (c) efforts to alleviate suffering. 4. Thank you for stating the following financial disclosure: CA - Comité Départemental De Lutte Contre Les Maladies Respiratoires, an independent association fighting for patient’s health Please state what role the funders took in the study.  If the funders had no role, please state: "The funders had no role in study design, data collection and analysis, decision to publish, or preparation of the manuscript." If this statement is not correct you must amend it as needed. Please include this amended Role of Funder statement in your cover letter; we will change the online submission form on your behalf. 5. We note that your Data Availability Statement is currently as follows: All relevant data are within the manuscript and its Supporting Information files. Please confirm at this time whether or not your submission contains all raw data required to replicate the results of your study. Authors must share the “minimal data set” for their submission. PLOS defines the minimal data set to consist of the data required to replicate all study findings reported in the article, as well as related metadata and methods (https://journals.plos.org/plosone/s/data-availability#loc-minimal-data-set-definition). For example, authors should submit the following data: - The values behind the means, standard deviations and other measures reported;- The values used to build graphs;- The points extracted from images for analysis. Authors do not need to submit their entire data set if only a portion of the data was used in the reported study. If your submission does not contain these data, please either upload them as Supporting Information files or deposit them to a stable, public repository and provide us with the relevant URLs, DOIs, or accession numbers. For a list of recommended repositories, please see https://journals.plos.org/plosone/s/recommended-repositories. If there are ethical or legal restrictions on sharing a de-identified data set, please explain them in detail (e.g., data contain potentially sensitive information, data are owned by a third-party organization, etc.) and who has imposed them (e.g., an ethics committee). Please also provide contact information for a data access committee, ethics committee, or other institutional body to which data requests may be sent. If data are owned by a third party, please indicate how others may request data access. Additional Editor Comments:

I apologize for the delay in handling your manuscript. We have gotten back reviews from three expert reviews in this field and they all conclude that the work reported is preliminary and more experiments are required to qualify your report. The manuscript needs heavy editing and addition of results from new experimental outcome. I am under the impression, it needs major revision if resubmitted.

Reviewers' comments:

Reviewer's Responses to Questions

**Comments to the Author**

1. Is the manuscript technically sound, and do the data support the conclusions?

Reviewer #1: Partly

Reviewer #2: No

Reviewer #3: Partly

2. Has the statistical analysis been performed appropriately and rigorously? 

Reviewer #1: Yes

Reviewer #2: Yes

Reviewer #3: I Don't Know

3. Have the authors made all data underlying the findings in their manuscript fully available?

Reviewer #1: Yes

Reviewer #2: No

Reviewer #3: Yes

4. Is the manuscript presented in an intelligible fashion and written in standard English?

Reviewer #1: Yes

Reviewer #2: No

Reviewer #3: Yes

5. Review Comments to the Author

**Reviewer #1:** Froment and colleagues determined the efficacy of mefloquine and its enantiometers using a murine model of pulmonary M.avium infection.

In the reported study, the investigators infected mice using an aerosol device for bacterial delivery and after 28 days treatment was initiated. The mice were treated with rifampin, clarithromycin, and ethambutol or clarithromycin, ethambutol and mefloquine (racemic mixture) or substituting the mefloquine in the regimen for + or – mefloquine enantiometer, daily, for 3 months.

The results suggest that the combination containing mefloquine (+) was the most active, although the results did not reach statistically significance.

Comments and Suggestions:

1. In the experimental design the group of animals that did not received treatment, was not described (methods section). Did they received similar amount of ethanol that was used to dissolve the compounds ? The description of methods can be improved.

2. The results obtained suggest that all treatments groups were equally effective. , which is not a surprise. The difference in CFUs is not significant, at months 1, 2 and 3 in the same treatment group and at a longitudinal comparison between groups. Did the authors tested for antibiotic resistance among the survival strains?

3. The number of animals used per time pint is quite small. I a couple of experimental groups one of the mouse was lost (the authors made no mention about the reason), which make the analysis difficult, in despite of the large difference in the number of CFU comparing treated and untreated mice.

4. The discussion of the manuscript needs to touch in many other points, regarding the disease and treatment. Mefloquine is used for the treatment of malaria. How it fits with the anti-bacterial action?

5. The legend of figure 1 needs to be improved, with better description of the experiments and definition of symbols.

**Reviewer #2: **The authors show, very succinctly, interesting results of an animal model of Mycobacterium avium infection and various 3 drug cocktails, including Mefloquine. Curiously, the cfu counts rebound over time, with the caveat that they had issues with the third month cultures. The authors show that they can do the drug treatments and successfully used an aerosol model of infection.

However, this very brief collection of preliminary results is essentially a promising pilot study. It includes a table that simply recapitulates the data shown in the graphs. Further, while this pilot study shows promise, the authors themselves admit to an issue with culturing the bacteria on the 3rd month. While not a disqualifying issue, it further emphasizes the need to replicate these result, including using larger groups of mice. We also would need more information on the pathology of lung, data on the cell subsets found, and, likely, followup for long term infections to see if the bacteria do become completely drug resistant. Adverse effects in the mice are not shown. There are numerous issue with grammar that need correction as well.

**Reviewer #3:** General comments

MFQ has been considered for the treatment of infections by M. avium and other mycobacteria. Studies by Bermudez et al. previously demonstrated that MFQ and its enantiomers (administered by gavage) have efficacy in mouse models of M. avium infection (refs 14 and 15 in the manuscript submitted by Froment et al. and PMID 12792877, which is not cited in the manuscript). Some of the work reported by Bermudez et al. included drug combinations, e.g., MFQ, EMB, and moxifloxacin in PMID: 12792877 or MFQ and EMB in ref 14. In these experiments, mice were infected intravenously (caudal vein), and bacterial loads (CFU) were quantitated in blood, liver, and spleen. However, bacterial loads in lung or lung gross pathology were not evaluated.

In the study submitted by Froment et al., the authors extend the investigation of MFQ and its enantiomers ((+)-EMQ and (-)-EMQ) efficacy using a model of M. avium pulmonary infection. They compared the efficacy of MFQ as a replacement for RIF in the CLR-EMB-RIF gold standard regimen. Their use of an aerosol model of M. avium pulmonary infection for treatment efficacy assessment is a significant contribution and contrasts with the previously reported models noted above. The replacement of RIF with MFQ in the CLR-EMB-RIF gold standard regimen is also novel. Based on their results, the authors conclude that MFQ and its enantiomers appear to be as effective as RIF in combination therapy.

The urgency for new and alternative treatments for M. avium infections cannot be overstated, making the subject of this study not only important but also highly relevant for PLOS ONE.

Major points

The conclusion is based on a single and relatively short experiment with three data points for bacterial loads in the lungs, one of which could have been compromised (month 3). Additional experiments are required to increase the robustness of the conclusion and to expand the significance of the contribution made by this manuscript. The authors should repeat the study. Extension of the study beyond three months and analysis of lung gross pathology should be added to the study.

What the bacterial loads in the lung would have been without RIF in the authors' experimental model is unknown. Is there a statistically significant difference between RIF-CLR-EMB and CLR-EMB in the authors' experimental model? If the answer is no, then MFQ and its enantiomers might have no impact on lung CFUs.

A CLR-EMB treatment group should be included in the study to address this uncertainty.

A question the authors could also address to expand their study's contribution is whether MFQ or its enantiomers alone impact lung CFUs. Is the pharmacokinetic of MFQ and its enantiomers in the mouse model used known? What concentration of MFQ and its enantiomers do the authors expect in lung tissue, and how do those concentrations relate to the MICs against M. avium?

Minor points

Does Table 1 show differences +/- SD or SEM? How were SD/SEM calculated?

How was enantiomeric purity assessed?

6. PLOS authors have the option to publish the peer review history of their article (what does this mean?). If published, this will include your full peer review and any attached files.

Reviewer #1: No

Reviewer #2: **Yes: **Alan R. Schenkel, Ph.D.

Reviewer #3: No

---

## [Author Response · Author response to Decision Letter 0]

23 May 2024

As indicated in the decision letter. All responses to reviewers are in the document named Response to Reviewers.

---

## [Decision Letter · Decision Letter 1]

9 Jul 2024

PONE-D-24-03260R1Efficacy of Mefloquine and its enantiomers in a murine model of Mycobacterium avium infectionPLOS ONE

Dear Dr. Froment,

Thank you for submitting your manuscript to PLOS ONE. After careful consideration, we feel that it has merit but does not fully meet PLOS ONE’s publication criteria as it currently stands. Therefore, we invite you to submit a revised version of the manuscript that addresses the points raised during the review process.

We look forward to receiving your revised manuscript.

Kind regards,

Delphi Chatterjee

Academic Editor

PLOS ONE

Journal Requirements:

Additional Editor Comments:

We sincerely apologize for this delay in sending feedback from the reviewers. For this R1 critiques, one reviewer felt substantial experimental more work was needed to qualify as a full length research paper. Another reviewer suggested accept as brief communication in this journal since the work was significant. I will go with the second reviewer's comments and perhaps shorten the length by 10% and submit as a brief communication

Reviewers' comments:

Reviewer's Responses to Questions

**Comments to the Author**

1. If the authors have adequately addressed your comments raised in a previous round of review and you feel that this manuscript is now acceptable for publication, you may indicate that here to bypass the “Comments to the Author” section, enter your conflict of interest statement in the “Confidential to Editor” section, and submit your "Accept" recommendation.

Reviewer #2: All comments have been addressed

Reviewer #3: (No Response)

2. Is the manuscript technically sound, and do the data support the conclusions?

Reviewer #2: Yes

Reviewer #3: Partly

3. Has the statistical analysis been performed appropriately and rigorously? 

Reviewer #2: Yes

Reviewer #3: I Don't Know

4. Have the authors made all data underlying the findings in their manuscript fully available?

Reviewer #2: Yes

Reviewer #3: Yes

5. Is the manuscript presented in an intelligible fashion and written in standard English?

Reviewer #2: Yes

Reviewer #3: Yes

6. Review Comments to the Author

Reviewer #2: This revised version of this Short Communication appears to have generally responded as much as needed to the prior critiques. There is precedent in this journal for the publication of similar brief studies, thus it doesn't appear necessary to the editors to ask for some of the major revisions asked for by the previous reviews.

Reviewer #3: The authors have not adequately addressed major points of concern. These concerns remain in the revised manuscript. The work still requires major revisions noted in the previous comments by this reviewer. The presented work is still viewed as a pilot study presenting interesting preliminary results that warrant adequate experimental follow-up and expansion to consolidate conclusions and provide a robust contribution to the field. I would encourage the authors to continue and expand this work to address major points of concern and resubmit their work for consideration.

7. PLOS authors have the option to publish the peer review history of their article (what does this mean?). If published, this will include your full peer review and any attached files.

Reviewer #2: **Yes: **Alan R. Schenkel, Ph.D.

Reviewer #3: No

---

## [Author Response · Author response to Decision Letter 1]

22 Aug 2024

- We agree to publish a brief communication

---

## [Editor Report · Decision Letter 2]

16 Sep 2024

Efficacy of Mefloquine and its enantiomers in a murine model of Mycobacterium avium infection

PONE-D-24-03260R2

Dear Dr. Froment,

We’re pleased to inform you that your manuscript has been judged scientifically suitable for publication and will be formally accepted for publication once it meets all outstanding technical requirements.

Kind regards,

Delphi Chatterjee

Academic Editor

PLOS ONE
---

## [Editor Report · Acceptance letter]

19 Sep 2024

PONE-D-24-03260R2 

PLOS ONE

Dear Dr. Froment, 

I'm pleased to inform you that your manuscript has been deemed suitable for publication in PLOS ONE. Congratulations! Your manuscript is now being handed over to our production team.

Kind regards, 

on behalf of

Dr. Delphi Chatterjee 

Academic Editor

PLOS ONE